# Epidemiological, Clinical and Genetic Features of ALS in the Last Decade: A Prospective Population-Based Study in the Emilia Romagna Region of Italy

**DOI:** 10.3390/biomedicines10040819

**Published:** 2022-03-31

**Authors:** Giulia Gianferrari, Ilaria Martinelli, Elisabetta Zucchi, Cecilia Simonini, Nicola Fini, Marco Vinceti, Salvatore Ferro, Annalisa Gessani, Elena Canali, Franco Valzania, Elisabetta Sette, Maura Pugliatti, Valeria Tugnoli, Lucia Zinno, Salvatore Stano, Mario Santangelo, Silvia De Pasqua, Emilio Terlizzi, Donata Guidetti, Doriana Medici, Fabrizio Salvi, Rocco Liguori, Veria Vacchiano, Mario Casmiro, Pietro Querzani, Marco Currò Dossi, Alberto Patuelli, Simonetta Morresi, Marco Longoni, Patrizia De Massis, Rita Rinaldi, Annamaria Borghi, Amedeo Amedei, Jessica Mandrioli

**Affiliations:** 1Department of Biomedical, Metabolic and Neural Sciences, University of Modena and Reggio Emilia, 41125 Modena, Italy; gianferrari.giulia@gmail.com (G.G.); marco.vinceti@unimore.it (M.V.); jmandrio@unimore.it (J.M.); 2Department of Neurosciences, Azienda Ospedaliero Universitaria di Modena, 41125 Modena, Italy; elibettizucchi@gmail.com (E.Z.); ceciliasimonini24@gmail.com (C.S.); fini.nicola@aou.mo.it (N.F.); gessani.annalisa@aou.mo.it (A.G.); 3Clinical and Experimental Medicine Ph.D. Program, University of Modena and Reggio Emilia, 41125 Modena, Italy; 4Department of Science of Public Health, Research Centre in Environmental, Genetic and Nutritional Epidemiology, University of Modena and Reggio Emilia, 41125 Modena, Italy; 5Department of Hospital Services, Emilia Romagna Regional Health Authority, 40127 Bologna, Italy; salvatore.ferro@regione.emilia-romagna.it; 6Neurology Unit, Arcispedale Santa Maria Nuova, AUSL-IRCCS Reggio Emilia, 42123 Reggio Emilia, Italy; elena.canali@asmn.re.it (E.C.); franco.valzania@asmn.re.it (F.V.); 7Department of Neuroscience and Rehabilitation, St. Anna Hospital, 44124 Ferrara, Italy; e.sette@ospfe.it (E.S.); pglmra@unife.it (M.P.); v.tugnoli@ospfe.it (V.T.); 8Department of Neuroscience and Rehabilitation, University of Ferrara, 44121 Ferrara, Italy; 9Department of Neuroscience, University of Parma, 43121 Parma, Italy; lucia.zinno@gmail.com (L.Z.); salvatore.stano89@gmail.com (S.S.); 10Department of Neurology, Carpi Hospital, 41014 Modena, Italy; m.santangelo@ausl.mo.it (M.S.); s.depasqua@ausl.mo.it (S.D.P.); 11Department of Neurology, G. Da Saliceto Hospital, 29121 Piacenza, Italy; e.terlizzi@ausl.pc.it (E.T.); d.guidetti@ausl.pc.it (D.G.); 12Department of Neurology, Fidenza Hospital, 43036 Parma, Italy; dmedici@ausl.pr.it; 13IRCCS Istituto delle Scienze Neurologiche di Bologna, Bellaria Hospital, 40139 Bologna, Italy; fabrizio.salvi@gmail.com; 14Dipartimento di Scienze Biomediche e Neuromotorie, University of Bologna and IRCCS Istituto delle Scienze Neurologiche di Bologna, Bellaria Hospital, 40139 Bologna, Italy; rocco.liguori@unibo.it (R.L.); veriavacchiano@gmail.com (V.V.); 15Department of Neurology, Faenza and Ravenna Hospital, 48121 Ravenna, Italy; mario.casmiro@auslromagna.it (M.C.); pietro.querzani@auslromagna.it (P.Q.); 16Department of Neurology, Infermi Hospital, 47923 Rimini, Italy; marco.currodossi@auslromagna.it (M.C.D.); marco.longoni@auslromagna.it (M.L.); 17Department of Neurology and Stroke Unit, “Morgagni-Pierantoni” Hospital, 47121 Forlì, Italy; alberto.patuelli@auslromagna.it; 18Department of Neurology and Stroke Unit, Bufalini Hospital, 47521 Cesena, Italy; simonetta.morresi@auslromagna.it; 19Department of Neurology, Imola Hospital, 40026 Bologna, Italy; p.demassis@ausl.imola.bo.it; 20IRCCS Istituto delle Scienze Neurologiche di Bologna, UOC Interaziendale Clinica Neurologica Metropolitana (NeuroMet), 40139 Bologna, Italy; rita.rinaldi@aosp.bo.it; 21IRCCS Istituto delle Scienze Neurologiche di Bologna, Department of Neurology and Stroke Center, Maggiore Hospital, 40133 Bologna, Italy; annamaria.borghi@isnb.it; 22Department of Experimental and Clinical Medicine, University of Florence, 50134 Florence, Italy; amedeo.amedei@unifi.it; 23SOD of Interdisciplinary Internal Medicine, Azienda Ospedaliera Universitaria Careggi (AOUC), 50134 Florence, Italy

**Keywords:** amyotrophic lateral sclerosis, epidemiology, population-based registry, incidence, clinical features, genetics

## Abstract

Increased incidence rates of amyotrophic lateral sclerosis (ALS) have been recently reported across various Western countries, although geographic and temporal variations in terms of incidence, clinical features and genetics are not fully elucidated. This study aimed to describe demographic, clinical feature and genotype–phenotype correlations of ALS cases over the last decade in the Emilia Romagna Region (ERR). From 2009 to 2019, our prospective population-based registry of ALS in the ERR of Northern Italy recorded 1613 patients receiving a diagnosis of ALS. The age- and sex-adjusted incidence rate was 3.13/100,000 population (M/F ratio: 1.21). The mean age at onset was 67.01 years; women, bulbar and respiratory phenotypes were associated with an older age, while C9orf72-mutated patients were generally younger. After peaking at 70–75 years, incidence rates, among women only, showed a bimodal distribution with a second slight increase after reaching 90 years of age. Familial cases comprised 12%, of which one quarter could be attributed to an ALS-related mutation. More than 70% of C9orf72-expanded patients had a family history of ALS/fronto-temporal dementia (FTD); 22.58% of patients with FTD at diagnosis had C9orf72 expansion (OR 6.34, *p* = 0.004). In addition to a high ALS incidence suggesting exhaustiveness of case ascertainment, this study highlights interesting phenotype–genotype correlations in the ALS population of ERR.

## 1. Introduction

From the recent global burden of disease studies, motor neuron diseases (MNDs), among which the most frequent is amyotrophic lateral sclerosis (ALS), has emerged as the fourth most common cause of death among neurological disorders in the USA, and they are among the disorders with the largest increase in absolute numbers in Western countries, mainly in age-standardized incidence-, mortality- and disability-adjusted life year rates [1]. Significant variations in incidence rates have been detected across different areas of the Western world, raising the need for the further exploration of epidemiological features of ALS in different countries, since different risk factors, policies and health care systems may have an impact on this disorder.

In addition, ALS is now considered a multisystem neurodegenerative disease, characterized by a heterogeneous clinical, pathologic and genetic background. In addition to motor symptoms, ALS patients may present cognitive and/or behavioral impairment, that in 15% of patients leads to the fulfilment of the diagnostic criteria for fronto-temporal dementia (FTD). Conversely, MND can appear during the course of FTD in up to 15% of patients [2].

Therefore, there is a clinical and pathophysiological continuum between FTD and ALS, demonstrated by the widespread pathological accumulation of TAR DNA-binding protein 43 (TDP-43) and some common genetic backgrounds, among which the C9ORF72 gene plays a key role [3].

Population-based registers represent the most reliable method of estimating the epidemiology of rare diseases such as ALS. As in other European and Western countries, in Italy, three regional registers (Piedmont and Valle d’Aosta, Liguria and Emilia Romagna, Northern Italy) for ALS have been collecting incident cases in recent years, and have documented incidence rates of 2–3/100,000 population [4,5,6]. Our previous epidemiological studies on ALS in the Emilia Romagna Region (ERR) allowed us to compute a crude ALS incidence of 2.63/100,000/year [4], with an increase over time of approximately 2%/year, with reference to the 2000–2009 period [7]. In this prospective population-based study, we aimed to identify the demographic and clinical features, the genotype–phenotype correlates and the temporal trends of ALS incident cases detected by the Emilia Romagna Registry (ERRALS) from 2009 to 2019.

## 2. Materials and Methods

### 2.1. Population

We established the prospective ALS registry of the ERR in 2009 [4], with the aim of collecting all incident ALS cases among ERR residents, following a validated disease diagnosis according to the revised El Escorial Criteria (EEC-R) [8]. The study was approved by the ethical committee of the coordinating and participating centers (Comitato Etico Provinciale di Modena, number 124/08, 2 September 2008). Caring physicians collected a detailed phenotypic profile of each ALS patient as previously described [4], uploading information about the patient into the registry database available through a dedicated website, with access restricted to the 17 ERR Neurological Departments. In each department, one or more investigators were identified as study referents, and could upload patients’ data at the time of diagnosis and information about the follow-up visits. Follow up of the patients was generally performed in all the ERR Neurological Departments until death, collecting, among other information, the following data: forced vital capacity (FVC); ALS Functional Rating Scale–Revised (ALSFRS-R) [9]; riluzole intake and discontinuation [10]; gastrostomy [11]; non-invasive (NIV) or invasive (IV) ventilation support; and cause, place and time of death. Systematic supervision and validation of the input data was performed by the coordinating center, checking for accuracy and completeness. Data from the population-based registry were integrated by cases discharged with the code 335.2 of the International Classification of Diseases (ICD 9th rev) or the corresponding G12.21 code of the ICD 10th revision, from regional hospitals, and by death certificates for residents reporting the same codes.

In this study, we focused on newly diagnosed ALS cases from 1 January 2009 to 31 December 2019, and we considered the following demographic and clinical variables: sex, age group, site of onset (bulbar; upper limb or lower limb; respiratory onset), clinical characteristics at neurological examination, ALSFRS-R and respiratory function (FVC), genetics and family history. Clinical phenotypes were classified as follows: bulbar ALS, classic ALS, flail arm ALS, flail leg ALS, Upper Motor Neuron predominant (UMNp) ALS and respiratory ALS, in line with the definition proposed by Chiò et al., 2020 [12]. The respiratory phenotype has to be distinguished from other ALS forms that develop respiratory insufficiency with advancing disease, and was defined as having prevalent respiratory involvement at onset, defined as orthopnea or dyspnea at rest or during exertion, with only mild spinal or bulbar signs in the first 6 months after onset [12].

### 2.2. Genetics

The registration of data in the genetic analysis included at least the presence/absence of SOD1, FUS, TARDBP mutations and C9ORF72 expansion, these four genes accounting for up to 70% of FALS [13].

The C9ORF72 status was determined by repeat primed PCR, as previously described (with individual laboratory-based validation and quality control using Southern blot analyses) [14]. All the coding exons and 50 bp of the flanking intron–exon boundaries of SOD1, of exon 6 of TARDBP and of exons 14 and 15 of FUS were PCR amplified, sequenced using the Big-Dye Terminator v3.1 sequencing kit (Applied Biosystems Inc., Foster City, CA, USA), and run on an ABIPrism 3130 genetic analyzer. These exons were selected as the vast majority of known pathogenic variants are known to lie within these mutational hotspots [14].

Depending on the results of genetic analyses and on the presence of family history of ALS and/or FTD, in a sub-group of patients, further genes were also explored using an NGS probe-based customized panel (Illumina Nextera Rapid Capture Custom kit, Illumina, San Diego, CA, USA) covering 78 genes, including ALS/FTD causative and susceptibility genes, hereditary motor neuronopathy (HMN) and hereditary spastic paraplegia (HSP) genes, as previously described [15].

### 2.3. Statistics

Continuous variables were reported as means (standard deviations, SD), and categorical variables as absolute numbers (relative frequencies as %). We evaluated clinical and demographic features according to gender, phenotype and genotype, performing 2-tailed t-tests and ANOVA for continuous variables between two groups or multiple groups, respectively, and chi-square tests for comparisons between categorical variables. A *p*-value < 0.05 was considered as statistically significant.

We grouped newly diagnosed patients by sex and five-year age categories, and computed incidence rates accordingly.

We reported crude and sex- and age-standardized incidence rates in the study populations. For the latter, we used direct standardization, using the 2011 national population as a standard. The number of unobserved cases was estimated according to the two-source capture–recapture method [16], using two independent sources of patient identification (ERRALS and hospital discharge archives). We computed the ninety-five percent confidence intervals (95%CIs) of the estimates assuming a Poisson distribution. Data analysis was performed using the STATA statistical package 15 (StataCorp. 2017. StataCorp LLC., College Station, TX, USA).

## 3. Results

### 3.1. Patients’ Clinical Features

From 1 January 2009 to 31 December 2019, 1613 residents of ERR received a new diagnosis of ALS, with a male to female ratio of 1.21, and a median age at onset and at diagnosis of 68.33 years (women: 68.92 years; men: 67.33 years), and 69.66 years (women: 70.25 years; men: 68.92 years), respectively. Table 1 shows the overall demographic and clinical features of the ALS patients, stratified by sex.

The progression rate at diagnosis increased with increasing age, with an ALSFRS-R total score monthly decline of 0.74 (±0.93) for patients <60 years at diagnosis, 0.94 (±0.93) for patients between 60 and 75 years at diagnosis and 1.48 (±1.79) for patients >75 years at diagnosis (*p* < 0.001). Consistently, the ALSFRS-R score was lower in older patients at diagnosis (40.67 ± 6.82 for patients aged <60 years, 38.46 ± 7.48 for patients aged 60–75 years and 35.82 ± 8.11 for patients aged >75 years; *p* < 0.001). Likewise, FVC (%) was lower in older patients at diagnosis (92.59 ± 24.04 for patients aged <60 years, 83.73 ± 24.66 for patients aged 60–75 years and 75.58 ± 27.79 for patients aged >75 years; *p* < 0.001). Table 2 shows patients’ demographics according to phenotypes.

### 3.2. Incidence and its Temporal Trend for the 2009–2019 Period

Based on 1613 newly diagnosed cases, the crude mean annual incidence rate was 3.30/100,000 (95%CI: 3.14–3.47), with an age- and sex-adjusted incidence rate for the 2011 Italian census population of 3.13/100,000. The overall ALS incidence was higher in men than in women (crude rates: 3.73/100,000 (95%CI: 3.49–3.99) vs. 2.90/100,000 (95%CI: 2.69–3.11), respectively; adjusted rates: 3.52/100,000 vs. 2.74/100,000, respectively).

The incidence increased with age from 40 with a peak at 70–74 years, followed by a sharp decrease (Figure 1). The crude incidence rate was higher in men than in women for each age group up to the age of 85, when an inversion of this ratio occurred. 

Incidence rates were generally stable during the years of the study. Considering the incidence rates in the two periods, 2009–2014 and 2015–2019, crude incidence rates increased from 3.59/100,000 (95%C.I. 3.27–3.94) in 2009–2014 to 3.90/100,000 (95%C.I. 3.54–4.29) in 2015–2019 among men and decreased from 3.13/100,000 (95%C.I. 2.84–3.44) to 2.62/100,000 (95%C.I. 2.33–2.94) among women (total incidence rates: 3.35/100,000 (95%C.I. 3.14–3.58) in the years 2009–2014 and 3.24 (95%C.I. 3.01–3.49) in the years 2015–2019). Using the two-source capture–recapture method, we estimated 68 unobserved cases with a coverage of 95.95%, which yielded an adjusted crude incidence rate of 3.44/100,000.

### 3.3. Patients’ Features According to Genetics

Genetic analysis was performed for 599 patients (37.13% of 1613 patients with newly diagnosed ALS). Among these, 523 (87.31%) patients did not exhibit gene mutations, 39 (6.51%) were C9orf72 expanded and 37 (6.18%) had other mutations, mainly represented by SOD1 (18 patients, 3.01%) and FUS (7 patients, 1.17%). 

Table 3 shows the demographic and clinical features of the genotyped patients.

Among the 201 ALS patients with a family history of ALS or FTD (fALS) (12.46% of the entire population), 36.39% carried a gene mutation related to ALS, whereas among the sporadic cases (sALS), only 6.53% were mutated/expanded (OR 8.16, *p* < 0.001). Among sALS, C9orf72 expansion and SOD1 mutation were equally represented (11 and 10 cases, 2.32% and 2.11%, respectively). Among fALS, C9orf72 expansion was far more frequent than all the other mutations (28 and 19 cases, 22.58% and 15.32%, respectively; OR 2.99, *p* < 0.02). Among patients with mutations/expansions in genes associated with ALS, an absence of family history was detected in 28.21% of patients with C9orf72 expansion, in 55.56% of patients with mutated SOD1, in 42.86% of patients with mutated FUS and in 58.33% of patients with mutated TARDBP.

Among the 28 patients with C9ORF72 expansion and family history, 12 had a family history of both ALS and FTD, nine had a family history of ALS and seven of FTD; all eight fALS patients with an SOD1 mutation had a family history of ALS, whereas no cases with a family history of FTD were detected. Among the four fALS with an FUS mutation, one had a family history of ALS and three of both ALS and FTD. As far as other gene mutations are concerned, two patients had a family history of ALS, two of FTD and one of both.

FTD at diagnosis was present in 62 ALS patients (10.35%) among those who underwent genetic analysis; 72.58% of patients with FTD at diagnosis did not carry genetic mutations, whereas 22.58% had C9orf72 expansion (OR 6.34, *p* = 0.004) and 1.61% FUS mutation. There were no patients with an SOD1 gene mutation among those presenting with FTD at diagnosis (*p* < 0.001).

ALS patients with C9orf72 expansion were the youngest at disease onset (56.68 ± 12.30 years; 64.08 ± 12.30 years for wild type patients, 63.08 ± 11.89 for SOD1 patients, 65.86 ± 13.79 for FUS patients, 66.67 ± 4.21 for TARDBP patients; *p* = 0.008), they had the shortest diagnostic delay (8.31 ± 5.74 months; 15.87 ± 17.42 months for wild type patients; 17.72 ± 15.81 for SOD1 patients; 20.43 ± 23.21 for FUS patients; 11.00 ± 1.41 for TARDBP patients; *p* = 0.127), and the lowest BMI at diagnosis (23.33 ± 4.32; 24.61 ± 3.96 for wild type patients, 25.27 ± 2.73 for SOD1 patients, 23.67 ± 4.07 for FUS patients, 26.00 ± 1.55 for TARDBP patients; *p* = 0.207). C9orf72-expanded patients had the fastest progression, as shown by the progression rate at diagnosis (−1.43 ± 1.42 points/month; −0.85 ± 0.98 points/month for wild type patients; −0.69 ± 0.69 for SOD1 patients; −0.67 ± 0.36 for FUS patients; −0.43 ± 0.49 for TARDBP patients; *p* = 0.025) and at last visit (−1.96 ± 1.45 points/month; −1.23 ± 1.50 points/month for wild type patients; −1.07 ± 1.35 for SOD1 patients; −1.44 ± 0.99 for FUS patients; −1.74 ± 1.64 for TARDBP patients; *p* = 0.123) and by respiratory decline, as measured by the FVC monthly decline (−5.73 ± 5.44% points/month; −2.57 ± 3.80% points/month for wild type patients; −2.86 ± 4.10 for SOD1 patients; −0.33 ± 1.86 for FUS patients; −3.39 ± 3.88 for TARDBP patients; *p* = 0.014).

## 4. Discussion

In this study, we report slightly higher ALS incidence rates than those already reported by European registers [4,5,6,7,8,9,10,11,12,16,17], which could be ascribed only partially to the age of the Emilia Romagna population, as the standardization to 2011 Italian census population confirmed incidence rates over 3/100,000 population—also higher than our previously reported estimates from 2000 onwards [7].

In the last decade, several studies on ALS in Western populations have shown an increase in ALS incidence [18], indicating that a global increase in the disease burden across the world in the next 25 years is likely to occur [19], and for which we cannot exclude the role of exposure to environmental factors such as increased air pollution, trauma, fractures and work-related exposure.

This has also been confirmed by the most longstanding registers in Europe (e.g., the Irish ALS register [20,21]) and in Italy (e.g., Piedmont and Aosta Valley Register for ALS (PARALS), located in northwestern Italy) [6].

Consistently with other studies, in our study, ALS incidence rates increased with older age [12,16,17,18,19,20,21,22] and peaked between 65 and 84 years, as has also been reported in France, Denmark and Sweden [23]. This well-known age-related incidence trend might be due to the depletion of susceptible individuals among the elderly, accumulation of exposure to risk factors (such as pesticides and other occupational and environmental risk factors [24,25,26,27]) exposure to triggering factors at an older age, or inter-disease competition [22].

We also observed an inversion of female incidence rates in the oldest age class, together with a flattening of incidence rates among men, which could be ascribed to data instability due to the small number of extremely old survivors [12,16,17,18,19,20,21,22,23,28]; however, a real increased incidence among very old women or a birth cohort effect [6] cannot be excluded. Furthermore, a role of under-ascertainment among the oldest old has already been hypothesized to explain the reduction in ALS age-specific incidence from 80 years old onwards, as these patients may be underdiagnosed due to comorbidities and reduced access to health services, but also due to a faster disease progression, as observed in our cohort.

The ERRALS population, at diagnosis, showed an older mean age at onset than previously reported [4,7], with a longer diagnostic delay and a faster disease progression [29]. The reason behind this distribution is not easy to explain; different exposure to environmental risk factors may play a role in this, but we could also argue that ERRALS includes a higher number of old and fast progressive patients at diagnosis, compared to other ALS cohorts [30]. Data collected for these patients from their first to last visit may have a relevant impact on the mean progression rate, which appears to be higher in ERRALS as compared to that reported elsewhere in literature and considered as a cut off value for patient stratification in clinical trials. This highlights that patients who are enrolled in clinical trials, as well as those followed in tertiary ALS Centers, may not really be representative of the general ALS population, as they are younger and have a slower disease progression [31].

Our age-adjusted incidence rates are among the highest reported in the literature, probably also reflecting the accuracy of case ascertainment, due to the distributed organization of ERRALS, in which all 17 regional neurological centers proactively participate [4], sharing ALS guidelines based on a multidisciplinary approach. This minimizes under-reporting from patients who have difficulty in travelling to distant tertiary ALS centers, and who are typically represented by faster progressors and older patients [31]. Additionally, the use of multiple sources for case ascertainment with an exhaustiveness greater than 95% achieved through the capture–recapture method may have played a key role, as pointed out elsewhere [23].

As far as the site of onset and phenotypes are concerned, among ERRALS patients, there is a higher frequency of classic, bulbar and respiratory phenotypes, compared to other population-based registers [32] where the patients presented quite frequently with more benign phenotypes, such as flail leg and UMNp phenotypes. This diversity may be due to true different ALS clinical presentations in the populations, but could also have been caused by methodological issues, related to the clinical classification and phenotype definition, which require a prolonged clinical follow up and, therefore, are operator dependent [32].

The mean diagnostic delay was unexpectedly higher than that found in previous studies in ERR [4], in Europe [17,33] and even in other continents [34]. This could be ascribed to the inclusion of less defined disease varieties inside the ALS spectrum definition, or to the inclusion of atypical cases [35], which come to the attention of neurological centers with greater difficulty, or later in time, and which have clinical complexity linked to other diseases [36,37]. Considering forthcoming personalized treatment, at least for genetic ALS, this diagnostic delay is far from being acceptable. As it relies mainly on medical choices, starting with the first doctor who sees the patient [38], it will be important to adopt strategies to counteract this latency, e.g., by alerting non-neurologists to some “red flag” symptoms that could suggest a serious neurological disorder [38] and should speed up the diagnostic process. Overall, the findings from our study suggest that the change in the burden of ALS in the Italian population is mostly related to the aging of the general population, as reported from different parts of the world [7,18,39], due to the improved survival rate of competitive diseases, better ALS ascertainment and increased awareness of disease heterogeneity, which may lead to the inclusion of less severe disease varieties, formerly considered as distinct diagnoses and now encompassed in the clinical spectrum of ALS.

In our ALS population, slightly more than 10% of all the patients reported a family history of ALS/FTD, whereas 37% of them were genotyped, showing a satisfactory diffusion of genetic testing in clinical practice with respect to other studies [34,40]. Consistent with previous reports, patients carrying genetic mutations were generally younger and had a shorter diagnostic latency [41]. C9orf72 patients presented a faster progression rate, while SOD1-related ALS showed an opposite, slow, progression. A higher prevalence of FTD in C9orf72 patients and in FUS was detected, as already reported [12,41,42,43]. A family history of ALS and FTD in C9orf72 patients was more frequent than in other genetic ALS, which is probably related to a higher penetrance of the gene [44,45]. Among C9orf72 patients, and among female patients, the bulbar phenotype was more frequent than in the general ALS population [46]. Moreover, C9orf72 patients had a lower BMI at diagnosis, which may be explained by the higher rate of the bulbar phenotype in this particular population, explaining the worse prognosis of this ALS patient subgroup [47], and may also indicate a faster disease progression before diagnosis.

With respect to other recent studies [48], we found a higher proportion of SOD1 patients with sporadic ALS, possibly due to some clusters of SOD1 in our region. On the other hand, the distribution of C9orf72-expanded patients across genetic fALS and sALS reflected previously reported findings (35.48% of genetic sALS versus 62.22% of genetic fALS) [48].

## 5. Conclusions

In a well-defined geographical region in Northern Italy, we observed incidence rates as high as those described in recent European population-based studies, pointing towards an increasing trend of ALS incidence mainly, but not only, due to the aging of the population. Consistently, ALS patients from ERRALS were older and had a fast disease progression, compared to other population-based registers [6,49], highlighting the exhaustiveness of the register, which also includes phenotypes that may be lost in patient cohorts collected mainly in tertiary centers. Genetic analysis is becoming increasingly important to better understand both the phenotypes and the underlying pathogenetic mechanisms, as well as finding possible personalized treatments for the disease. From this perspective, describing the natural history of disease associated with genetic mutations is noteworthy and deserves further biological investigations including emerging prognostic biomarkers [50,51,52].The main strength of this study lies in the use of multiple sources for data collection, with a special focus on a population-based registry involving a widespread network of ALS neurologists in ERR as the primary source of cases, the length of follow up and the high incidence rate, which reflects the high accuracy and exhaustiveness of case ascertainment. Some limitations must also be acknowledged and are mainly represented by missing data on clinical phenotypes, and, in particular, on cognitive/behavioral impairment details (onset and impaired domains) and disease progression rates, as measured by ALSFRS-R and FVC decline.

## Figures and Tables

**Figure 1 biomedicines-10-00819-f001:**
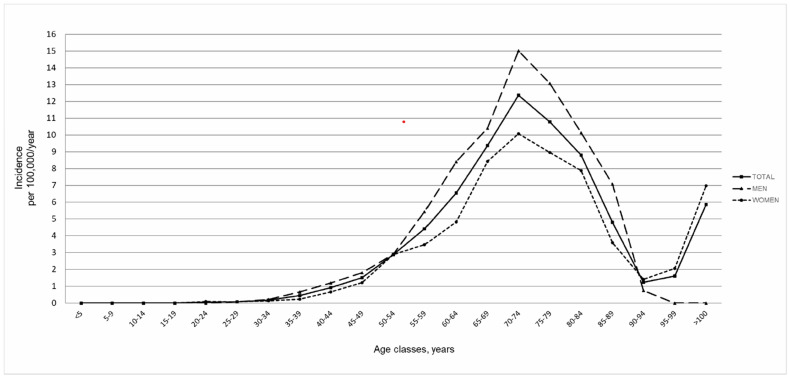
Incidence rates for 5–year age classes in males and females.

**Table 1 biomedicines-10-00819-t001:** Demographic and clinical features of the ALS patients, stratified by sex.

Demographic and Clinical Features	Women N (%), Mean [SD]	Men N (%), Mean [SD]	Total N (%), Mean [SD]	*p*-Value
Mean age at onset, years	67.84 [11.52]	66.32 [11.34]	67.01 [11.44]	0.009
Mean age at diagnosis, years	68.95 [11.47]	67.33 [11.33]	68.06 [11.42]	0.004
Mean diagnostic delay, months	13.57 [12.67]	13.74 [14.05]	13.67 [13.44]	0.807
Site of onset				<0.001
Bulbar	239 (32.83)	219 (24.75)	458 (28.30)	<0.001
Spinal, UL	146 (20.05)	268 (30.28)	414 (25.67)	<0.001
Spinal, LL	182 (25.00)	226 (25.54)	408 (25.29)	0.805
Respiratory	10 (1.37)	20 (2.26)	30 (1.86)	0.189
Phenotype				<0.001
Bulbar	237 (32.55)	218 (24.63)	455 (28.20)	<0.001
Classic	220 (30.21)	347 (39.21)	567 (35.15)	<0.001
Flail arm	24 (3.30)	39 (4.41)	63 (3.91)	0.252
Flail leg	45 (6.18)	75 (8.47)	120 (7.44)	0.081
UMN-p	42 (5.77)	37 (4.18)	79 (4.90)	0.141
Respiratory	10 (1.37)	20 (2.26)	30 (1.86)	0.190
FTD	49 (6.73)	64 (7.23)	113 (7.01)	0.829
Family history of ALS/FTD	84 (11.53)	117 (13.22)	201 (12.46)	0.433
ALSFRS-R score at diagnosis	37.39 [8.19]	39.27 [7.09]	38.45 [7.65]	<0.001
Progression rate at diagnosis *	1.13 [142]	0.92 [1.00]	1.01 [1.21]	0.002
Progression rate—first to last visit ^#^	1.38 [1.96]	1.41 [1.73]	1.39 [1.83]	0.786
FVC at diagnosis	84.61 [27.36]	84.66 [24.77]	84.64 [25.85]	0.979
Monthly FVC decline ^§^	2.84 [4.35]	2.53 [3.60]	2.65 [3.92]	0.344
Riluzole treatment	479 (65.80)	643 (72.66)	1122 (69.56)	0.009
Death	566 (77.74)	682 (77.06)	1248 (77.37)	0.638
Total	728 (45.13)	885 (54.87)	1613 (100)	

Table 1 shows differential distribution of main clinical features between sexes tested with a chi-square test or t-test as appropriate. Spinal UL: spinal upper limb; Spinal LL: spinal lower limb; UMN-p: Upper Motor Neuron predominant; FTD: frontotemporal dementia; FVC: forced vital capacity; SD: standard deviation. * Progression rate at diagnosis was calculated as monthly decline in ALSFRS-R score assuming a total score of 48 at onset; it was calculated for 1162 patients. ^#^ Progression rate from first to last visit was calculated as monthly decline in ALSFRS-R score from the first to the last available visit or from the first visit to the first time when a total score of 0 was recorded (whichever came first). It was calculated for 1049 patients in total. ^§^ Monthly decline in FVC was calculated for 600 patients in total, considering first FVC% value, the last available FVC% value and time between the first and last FVC.

**Table 2 biomedicines-10-00819-t002:** Characteristics of patients according to phenotypes.

Phenotype	Bulbar, N (%), Mean [SD]	Classic, N (%), Mean [SD]	Flail Arm, N (%), Mean [SD]	Flail Leg, N (%), Mean [SD]	UMNp, N (%), Mean [SD]	Respiratory, N (%), Mean [SD]	Unknown, N (%), Mean [SD]	*p*-Value
Mean age at onset, years	69.84 [10.69]	64.84 [11.38]	65.01 [10.53]	65.67 [11.47]	61.57 [12.68]	70.35 [6.08]	67.50 [10.97]	<0.001
Mean age at diagnosis, years	70.74 [10.70]	66.05 [11.44]	66.38 [10.40]	67.11 [11.16]	63.41 [12.75]	71.30 [6.24]	68.52 [10.87]	<0.001
ALSFRS-R score at diagnosis	38.31 [7.71]	38.58 [7.52]	38.69 [8.57]	38.94 [6.72]	39.06 [6.99]	34.07 [9.30]	36.00 [18.07]	0.095
Progression rate at diagnosis *	1.27 [1.57]	0.92 [0.98]	0.67 [0.72]	0.68 [0.65]	0.75 [0.78]	1.67 [1.65]	0.61 [0.45]	<0.001
Progression rate, first to last visit ^#^	1.55 [2.08]	1.36 [1.67]	1.33 [1.51]	1.25 [1.71]	0.87 [1.08]	1.74 [2.60]	1.58 [2.64]	0.151
FVC at diagnosis	77.15 [25.83]	88.78 [25.59]	92.04 [23.84]	88.03 [23.23]	95.11 [21.48]	62.05 [17.29]	83.00 [14.14]	<0.001
Monthly FVC decline ^§^	2.89 [4.53]	2.82 [3.69]	2.25 [3.96]	2.17 [3.54]	2.14 [3.34]	0.46 [3.11]	3.36 [3.44]	0.323
FTD	50 (10.99)	47 (8.29)	5 (7.94)	2 (1.67)	6 (7.59)	1 (3.33)	2 (0.74)	0.032 **
Family history of ALS/FTD	61 (13.41)	99 (17.46)	12 (19.05)	13 (10.83)	5 (6.33)	4 (13.33)	7 (2.58)	0.089 ***
Total	455 (28.20)	567 (35.15)	63 (3.91)	120 (7.44)	79 (4.90)	30 (1.86)	271 (16.80)	

Table 2 shows differential distribution of main clinical features among phenotypes, tested with chi-square test or ANOVA as appropriate. FVC: forced vital capacity; FTD: frontotemporal dementia; UMN-p: Upper Motor Neuron predominant. * Progression rate at diagnosis was calculated as monthly decline in ALSFRS-R score assuming a total score of 48 at onset; it was calculated for 1159 patients in total. ^#^ Progression rate from first to last visit was calculated as monthly decline in ALSFRS-R score from the first to the last available visit or from the first visit to the first time when a total score of 0 was recorded (whichever came first). It was calculated for 1049 patients in total. ^§^ Monthly decline in FVC was calculated for 600 patients in total, considering first FVC% value, the last available FVC% value and time between the first and last FVC. ** single comparisons showed a different frequency of FTD between classic and flail leg (*p* = 0.010), flail arm and flail leg (*p* = 0.032), flail leg and UMNp (*p* = 0.037) phenotypes; *** single comparisons among phenotypes showed a different frequency of family history between classic and UMNp (*p* = 0.012), flail arm and UMNp (*p* = 0.020) phenotypes.

**Table 3 biomedicines-10-00819-t003:** Demographic and clinical features of genotyped patients.

Genotype	No Mutations, N (%)	C9ORF72, N (%)	SOD1, N (%)	FUS, N (%)	Other Genes, N (%)	*p*-Value
Sex, men	305 (58.32)	17 (43.59)	11(61.11)	5 (71.43)	9 (75.00)	0.374
FTD	45 (8.60)	14 (35.90)	0 (0.00)	1(14.29)	2 (16.67)	<0.001 *
Family history of ALS/FTD	79 (15.11)	28 (71.79)	8 (44.44)	4 (57.14)	5 (41.67)	<0.001 **
Site of onset						0.097
Bulbar	169 (32.31)	12 (30.77)	3 (16.67)	2 (28.57)	1 (8.33)	0.285
Spinal UL	167 (31.93)	11 (28.21)	4 (22.22)	4 (57.14)	7 (58.33)	0.143
Spinal LL	161 (30.78)	14 (35.90)	10 (55.56)	1 (14.29)	2 (16.67)	0.112
Respiratory	13 (2.49)	0 (0.00)	0 (0.00)	0 (0.00)	2 (16.67)	0.021
Phenotypes						0.229
Bulbar	170 (32.50)	12 (30.77)	3 (16.67)	2 (28.57)	1 (8.33)	0.276
Classic	232 (44.36)	21 (53.85)	9 (50.00)	4 (57.14)	10 (83.33)	0.068
Flail arm	25 (4.79)	3 (7.69)	0 (0.00)	1 (14.29)	0 (0.00)	0.467
Flail leg	37 (7.07)	1 (2.56)	4 (22.22)	0 (0.00)	0 (0.00)	0.061
UMNp	31 (5.93)	0 (0.00)	1 (5.56)	0 (0.00)	0 (0.00)	0.459
Respiratory	14 (2.68)	2 (5.13)	1 (5.56)	0 (0.00)	0 (0.00)	0.767
Total	523 (87.31)	39 (6.51)	18 (3.01)	7 (1.17)	12 (2.00)	

Table 3 shows differential distribution of main clinical features among genotypes tested with the chi-square test. Spinal UL: spinal upper limb; Spinal LL: spinal lower limb; UMN-p: Upper Motor Neuron predominant; FTD: frontotemporal dementia; SD: standard deviation. * single comparisons showed a different frequency of FTD between patients without gene mutation and patients carrying C9ORF72 expansion (*p* < 0.001), and between patients carrying SOD1 mutations and C9ORF72 expanded patients (*p* = 0.003). ** single comparisons showed a different frequency of family history between patients without gene mutation and patients carrying C9ORF72 expansion (*p* < 0.001), between patients without gene mutation and patients carrying SOD1 mutations (*p* = 0.001), between patients without gene mutation and patients carrying FUS mutations (*p* = 0.002) and between patients without gene mutation and patients carrying mutations in other genes (*p* = 0.012); furthermore a different family history frequency was detected between SOD1 and C9ORF72 patients (*p* = 0.047).

## Data Availability

Data are available from the authors upon reasonable request.

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
