# Peer review of "Epidemiological, Clinical and Genetic Features of ALS in the Last Decade: A Prospective Population-Based Study in the Emilia Romagna Region of Italy"

_biomedicines, 2022, doi:10.3390/biomedicines10040819_

Round 1
Reviewer 1 Report
The manuscript submitted by Gianferrari et al. describes epidemiological, clinical and genetic features of ALS in one area of Italy. The population is not so large. However, the results of this paper based on up-to-date data will be necessary for the understanding of the trend of incidence rates. I have a comment on genetic analysis as follows:
The details on the genetic analysis system in Italy should be described. Did one institution perform the gene analysis of samples from patients in Emilia Romagna Region by using a gene panel? If the gene panel was used for the testing, the authors should list the examined genes.
Reviewer 2 Report
Dear Authors,
Appreciating for your long term effort to generate this kind of meaningful and valuable data for society.
however there is no such comments from my side except:
please correct the sentence which is negatively sounds "We collapsed patients diagnosed by sex and five-years age categories, and we
computed incidence rates accordingly" and also look into the whole manuscript for appropriate spacing between the sentences.
Reviewer 3 Report
Interesting data on the frequency of ALS in one of the regions of Italy, but there are a number of questions.
1. Introduction, first phrase - the authors use "and" between MND and ALS. But ALS is a form of motor neuron disease, along with PLS and SMA. Provided that the authors actually analyzed only ALS - you can make changes. And it is also important to add information to the introduction. according to FTD as a disease related in its genetic nature to ALS.
2. Unsuccessful format of tables 1-3. It is not clear what the value of H refers to in blitz 3 - parts of the rows of the table. It is not clear what one value of n refers to in a row of a table with 5 columns with different data. What is compared with what is not spelled out in the tables.
3. Table 3 - ALS and FTD. What was primary in the diagnosis - motor disorders or dementia? Is it possible to divide the family history into pure ALS, into cases with only FTD until the first case of ALS, into mixed FTD\ALS.
4. Results, 3.1, third paragraph. Again the question of what r. And how did you manage to get such values ​​of n with a very strong overlap of ASLFRS-R score between different groups?
Reviewer 4 Report
Gianferrari and several physicians that follow ALS cases established a registry for ALS from 2009 to 2019, and the present study deals with a retrospective population-based registry of ALS in the Emilia Romagna Region of Northern Italy that counts 1613 patients receiving a diagnosis of ALS.
Age and sex-adjusted incidence rate was 3.13/100,000 population (M/F ratio:1.21). Mean age at onset was 67 years; women, bulbar and respiratory phenotypes were associated with older age, while C9orf72-mutated patients were generally younger in age, further a subtype with a "respiratory ALS subtype" is presented according to Chio definition, (Neurology 2020) since most ALS cases have respiratory involvement a clinical definition could be added.
Results show higher prevalence in male cases, progression rate at diagnosis with a major decline for patients over 60 years old, ALSFR-R score lower in older ALS cases.
The overall patient prevalence is higher than in other studies in Italy (Cima et al.Eur J.Neurol.2009;16(8):920-4.doi: 10.1111/j.1468-1331.2009.02623.) that accounted for 1.22 to 1.90 cases/100,000 populations one wonders if this is due to local exposure to risk factors such as trauma or pesticides or the observations in a rather aging population.
ALS incidence rate peaked between 64 and 85, with a sharp decrease but in that age group in Fig.1 an inversion is then observed, but not discussed.
Table 3 for genetic ALS with C9ORF72 mutation include both FTD and ALS/FTD are grouped if among patients with this mutation such inclusion criteria are used was this a possible cause of higher incidence of ALS in this study? Inclusion criteria to exclude AD and MCI patients need clarification. In this study 70% of C9orf72-expanded patients had a family history for ALS/frontotemporal dementia(FTD); 23 % of patients with FTD at diagnosis had C9orf72 expansion.
ALS is a fatal neurodegenerative disease characterized by the neurodegeneration of motoneurons. About 10% of ALS are hereditary and involve a mutation in 25 different genes, while 90% of the cases are sporadic forms of ALS. The diagnosis of ALS includes the detection of early symptoms and, as the disease progresses, muscle twitching and then atrophy spreads from hands to other parts of the body. The disease causes high disability and has a high mortality rate; moreover, the therapeutic approaches for the pathology are not effective, although rehabilitation and prescribed exercise might have a role, was this considered in the current study prognosis?.
miRNAs are small non-coding RNAs, whose activity has a major impact on the expression levels of coding mRNA. The literature identifies several miRNAs with diagnostic abilities on sALS, a unique diagnostic profile is defined for C9orf72 mutated cases see The role of prescribed exercise in ALS July 2021 in Expert Review of Neurotherapeutics is covered.
As miRNAs could be secreted, the identification of specific blood miRNAs with the diagnostic ability for sALS could be helpful in the identification of the patient subtype, Authors could comment on biomarkers studied in screened cases.
Round 2
Reviewer 3 Report
The question remains on tables 1 and 3. What is the p value <0.001 related to, what exactly was compared and how. Table 2 by 5 and criterion chi 2 ? This needs to be clearly spelled out - it is not yet possible to understand what p refers to.
Reviewer 4 Report
The still pending issue in present study is the rather high prevalence and incidence of ALS .Authors state they cannot exclude that exposure to environmental factors may contribute to the previously reported increasing trend in ALS incidence and prevalence . On this issue, could they reasonably exclude other factors such as increased air pollution, trauma, fractures and work related exposure.
